# Application of Precision-Cut Lung Slices as an In Vitro Model for Research of Inflammatory Respiratory Diseases

**DOI:** 10.3390/bioengineering9120767

**Published:** 2022-12-04

**Authors:** Yan Liu, Ping Wu, Yin Wang, Yansong Liu, Hongfang Yang, Guohua Zhou, Xiaoqi Wu, Qingping Wen

**Affiliations:** 1Anesthesiology Department, Dalian Medical University, Dalian 116041, China; 2Department of Anesthesiology, The First Affiliated Hospital of Dalian Medical University, Dalian 116014, China; 3Department of Anesthesiology, Dalian University Affiliated Xinhua Hospital, Dalian 116021, China; 4Ningbo First Hospital, Ningbo 315016, China

**Keywords:** precision-cut lung slices, lung inflammation, ex vivo lung models, respiratory system, human lung tissue

## Abstract

The leading cause of many respiratory diseases is an ongoing and progressive inflammatory response. Traditionally, inflammatory lung diseases were studied primarily through animal models, cell cultures, and organoids. These technologies have certain limitations, despite their great contributions to the study of respiratory diseases. Precision-cut lung slices (PCLS) are thin, uniform tissue slices made from human or animal lung tissue and are widely used extensively both nationally and internationally as an in vitro organotypic model. Human lung slices bridge the gap between in vivo and in vitro models, and they can replicate the living lung environment well while preserving the lungs’ basic structures, such as their primitive cells and trachea. However, there is no perfect model that can completely replace the structure of the human lung, and there is still a long way to go in the research of lung slice technology. This review details and analyzes the strengths and weaknesses of precision lung slices as an in vitro model for exploring respiratory diseases associated with inflammation, as well as recent advances in this field.

## 1. Introduction

Respiratory diseases are often caused by an imbalance in the body’s ability to control inflammatory responses [1]. The most common inflammatory lung diseases include pathogen-infected pneumonia, acute lung injury (ALI), and chronic obstructive pulmonary disease (COPD). Although many disease models have been widely reported for use in studying the mechanisms of inflammation, there are many limitations in their applications, and it is, therefore, necessary to find a disease research model that can be fully modeled in vitro as soon as possible [2].

In previous studies, the research models on inflammatory respiratory diseases mainly relied on alternative animal models and cell culture techniques. Traditional animal models can directly study the effects of inflammation on the respiratory system in the physiological environment of animals, which is an important tool used to explore the pathogenesis of pulmonary inflammatory diseases [3]. However, due to the differences between species, the converted research results cannot accurately reflect the inflammatory pathogenesis of human lung diseases, which limits the further application of animal models. A two-dimensional (2D) in vitro culture of primary human lung cells is more of the original lung tissue’s morphological and physiological characteristics, but due to the lack of spatial structure, the findings of these models are still speculative in their assessment of the degree of inflammation of the primary lung, lacking accuracy and credibility [4]. From the perspective of 3D models, the air-liquid interface (ALI) can better resolve the spatial structures of the lungs and can fully reproduce the differentiation of human natural gas duct epithelial tissues exposed to pathogenic factors in vitro, which marks an improvement over 2D cultures. However, the resident cells of the ALI airway epithelial model lack mesenchymal cells, immune cells, and endothelial cells, among others. These cells, in turn, play a key regulatory role in the inflammatory response and in the immune response within the airway stimulated by exposure factors and, thus, require additional supplementation to reproduce the inflammatory response in vivo [5,6].

Recently, models such as organoids and microfluidic lung chips were developed that are more similar to the structure of human lungs [7]. “Pulmonary organoids” can be isolated and amplified from different areas of the mouse and human lungs and then inoculated into a 3D culture to promote self-organization. Transformed human lung cell lines can also be used in these models, which are an important tool in basic and regenerative medicine and other research fields. However, pulmonary organoids cannot recreate all of the complex structures and cellular interactions in different areas of the lungs, especially the fragile alveoli, which are rich in blood vessels, or the microenvironment of cells in the body [8]. Moreover, a transformed human cell line is already different from the human lung cells themselves. Although pulmonary organoids can simulate the stretching movement of lung tissue during respiration well, they are relatively complicated to model due to their dependence on artificial scaffolds, and it is also very difficult to obtain primary cells from patients, which limits the development of this model [9]. Most microfluidic lung chips consist of the outer portions of two microchannels that can be separated into gas–blood cavities and attached vacuum channels using a polydimethylsiloxane (PDMS) membrane to simulates changes in lung motion and lung function at the organ level. The microenvironment in vivo was restored by analyzing the recruitment of circulating immune cells under living blood flow by manipulating the fluid in a tube tens of microns in size [10]. Microfluidic lung chips can preserve the key structure of the human alveolar–capillary interface well and accurately reflect tissue–tissue interfaces, epithelial–endothelial crosstalk, and immune cell–host response [11]. However, even this is not fully consistent with the changes in barrier thickness, pressure, and flow in the natural alveolar capillary barrier in vivo. The more obvious deficiency is the lack of alveolar macrophages, which still cannot preserve the various types of cells and extracellular matrix associated with inflammatory diseases in the lungs [12]. Bioreactors are the most critical element in lung bioengineering and can be seen as vessels for in vitro or ex vivo studies and for the growth of cell and tissue systems, which are more physiologically relevant than traditional static in vitro models [13].Fluid systems composed of bioreactors allow cells and tissues to be dynamically cultured in a more similar physiological environment of air and fluid flow and can also be used to assess the effects of shear stresses, bacterial infections, and drugs on cells [14]. However, the lung flow model under such dynamic conditions does not have uniform velocity parameters at present, and the cell density of different cells in culture is also different. The wall shear stress provided by the fluid system is high, and most cells are affected by the edge effect and located in the periphery of the system, lacking interaction with other cells, which is different from in vivo and needs to be further solved [15]. Therefore, it is necessary to find a more reductive model of the lung environment to promote the research of lung diseases.

Precision-cut lung slices (PCLS) are thin and uniform 3D lung slices made from human or animal lung tissues that have attracted increasing attention both at home and abroad [9]. PCLSs derived from human or other animal lung tissues are special in that they can reduce the cellular structure and extracellular matrix (ECM) in the native lung environment well while preserving the interaction between different cell types and the ECM, which is difficult to accomplish in other models. At the same time, using the PCLSs of patients with different diseases, more targeted research on various diseases can be carried out, and differentiated treatment plans can be applied to patients based on these models. PCLSs are expected to become an indispensable tool for basic and applied lung research (Figure 1).

## 2. Preparation and Storage of PCLSs

Tissue sectioning techniques were first used in the 1920s [16]. It is challenging to make a lung section because of the special structure of the soft alveolar tissue. In a breakthrough study by Placke et al., an agarose perfusion method was used to successfully support the soft honeycomb structure of the lung, opening up a new era of using in vitro model lung slices to study diseases [17]. With the advent of the Krumdieck Slicer, the Vibratome, it become possible to cut agar-filled lungs into a large number of tissue slices with greater precision, reproducible thicknesses, and less tissue damage [16]. PCLSs began to be widely used as in vitro organ section culture systems and played an important role in the study of respiratory diseases [18,19,20].

Most experimental studies were conducted on rodent PCLSs, but the preparation methods of lung sections vary according to the size of the animal and the space required for research methods in terms of agarose concentration, the thickness of lung sections, and cell medium, etc.; so far, no uniform preparation standard has been established [21]. The most common method used today is to gently fill the lungs with an appropriate volume of warm agarose solution with a concentration of 0.5–3%, depending on the size of the lungs, to allow them to fully expand. The lungs are then removed from the chest cavity and cooled on ice for 10 min. They are then stored in a pre-cooled DMEM/F-12 medium at 4 °C for 20 min until the agarose solidifies. A biopsy punch is used to generate a tissue cylinder with a diameter of 8 mm. Then, an automatic microtome is used to perform precise cutting, usually with a thickness of 100–500 μm. For incubation purposes, 50 μg/mL gentamicin and 0.25 μg/mL amphotericin B are added to serum-free minimal element medium (MEM) and then placed in a 5% CO_2_ incubator at 37 °C. The medium is changed every 30 min for the first 2 h, 1 h/time for the next 2 h, and then switched to 24 h/time to thoroughly remove the residual agarose and cell debris (Figure 2) [22].

Human lung sections (hPCLS) were found to be more translatable and advantageous in the study of lung diseases. hPCLSs can be obtained from diseased lung lobes surgically removed from healthy and diseased human lung tissue and can be replicated multiple times, selecting specific regional tissues to simulate disease and accurately evaluate treatment outcomes [21]. Unlike animal PCLSs, hPCLSs are usually infused through the bronchus with 1.5–2% warm agaroses at low temperatures. After pathologically ruling out the risk of tumor and infection, hPCLSs are prepared at low temperature using a tissue biopsy machine, usually with a thickness of 200–300 μm and a diameter of approximately 8 mm [23]. If there is no bronchus in the lung tissue, agarose can be randomly inserted into the alveolar space tissue or directly injected into the alveolar cavity to fully expand the lung structure while preserving it, and then, tissue sections can be prepared [24,25].

In terms of the long-term preservation of PCLSs, since human lung tissue specimens are very rare, and lung samples from diseased patients are even more so, a good storage scheme is needed to preserve a large amount of PCLSs for subsequent use. To prolong the storage time of PCLSs, we suspended PCLS in standard medium DMEM/F-12 and then added 10% dimethyl sulfoxide (DMSO) for cryopreservation at −80 °C [26,27,28]. Finally, it was found that cryopreservation had almost no negative effect on the activity and proliferation of immune cells such as phagocytes and T lymphocytes. Surprisingly, the airway contraction and relaxation abilities of specific agonists and antagonists were still retained in thawed PCLSs, and the contractile force of smooth muscle cells in the airway was still well regulated by Ca^2+^-dependent mechanisms [27,28]. In order to optimize the refrigeration method of PCLSs, Tigges et al. refrigerated the standard medium DMEM/F-12 for 14 days after adding a high concentration of potassium chloride; they found that the response of PCLSs to inflammatory stimuli and their bronchial constriction ability were significantly improved compared with PCLSs cultured in DMEM/F-12 standard medium only [29]. In terms of PCLS cultures, due to the complex mechanisms of chronic lung disease, the use of PCLSs may be required in longer studies to observe irreversible ECM remodeling. Khan et al. found that with the increase in in vitro culture time, the proteomic and metabolomic levels of hPCLS from pulmonary fibrosis patients would change over time. The expression of proteins related to ECM degradation and inflammatory signaling pathways were significantly up-regulated [24]. The mesenchymal cells, epithelial cells, endothelial cells, and immune cells that make up human lung tissue in hPCLS of most patients with pulmonary fibrosis can be retained for two weeks without the addition of fetal calf serum (FCS), and the vascular endothelial cells can be shed in the first week [24,30]. Studies have shown that the addition of FCS to the culture medium did not increase the cell activity in rat PCLSs, and the incubation time in serum-free conditions was a breakthrough at 29 days [31]. The above studies suggest that finding a standardized and uniform PCLS medium and continuously improving it is an effective way to extend the culture time of PCLSs, but further exploration is still needed. Hydrogel biomaterials are ideal for simulating the ECM. The latest study found that embedding PCLSs in hydrogel can extend the in vitro culture time from the original 2 weeks to 21 days and maintain the dry function of epithelial cells. This method provides more opportunities for the application of PCLSs in the study of chronic lung diseases [32,33]. The study of synthetic scaffold matrices to support the growth of PCLSs may be an important direction to optimize preservation methods.

## 3. Combined Application of PCLSs and Modern Technology

With the progress of science and technology, new opportunities emerged to combine PCLS technology with other modern technologies in order to optimize the application of PCLSs in disease research, setting off a new wave of PCLS model applications. For example, it was found that the contraction in the peripheral airway could be observed when video phase contrast microscopy was applied to human or mouse lung sections [34]. Alveoli, which are used for gas exchange, are one of the major lesion sites in lung diseases [35]. Moreover, in the process of alveolar development, insufficient alveolar formation is an important cause of bronchopulmonary dysplasia (BPD) in premature infants. The alveolar structure can be well preserved in PCLS models. Time-lapse imaging technology can be used to display the evolution of alveolar development in real time. Through image tracking, specific dynamic behaviors of epithelial cells that contribute to alveolar formation, such as cell aggregation, hollowing, and cell extension, can be observed [35]. Branchfield et al. combined gene markers, immunofluorescence staining, confocal optical sectionals, and 3D reconstruction to perform visual analyses of myofibroblasts, alveolar epithelial cells, and other cells in the process of alveolar formation in terms of cell structure and the spatial relationships between cells in a PCLS model [36]. A recent study has made good use of PCLSs to investigate the role of macrophage-derived IL-6 in the pathogenesis of bronchopulmonary dysplasia in neonates. This also suggests that the effective combination of PCLS and cytokine pretreatment can be better applied in research [37]. As a high-resolution, real-time imaging platform, PCLSs can be used to evaluate the progression of pathogenic infection and inform adjustments to medication by applying fluorescent markers and probes to capture the ultrastructural changes and signal transductions in different cell membranes in real time. The molecular mechanism of cells in PCLS can be detected with small molecule inhibitors, which greatly reduces research costs and reduces drug usage compared with in vivo experiments [24,38]. Confocal laser scanning microscopes (CLSM) can be used in the immunofluorescence labeling of antibodies and cell-specific markers of bacterial or viral proteins to detect typical bacterial or viral inclusion bodies in a short time after infection, allowing researchers to explore the effects and mechanisms of bacterial or viral infection on lung tissue in vivo [39]. Video microscopy in PCLSs to study the influence of changes in airway structure on airway reactivity is effective, and even the smallest airway can be visualized to track its contraction [40]. When examining cell viability on PCLSs, silver nanoparticles were found to promote collagen I synthesis, as well as the release of inflammatory cascade cytokines, indicating a proinflammatory effect of nanomaterials [41]. In order to better restore the effect of mechanical and physical stimulation generated by lung stretching movement on the lung in vitro model, Dassow et al. mounted a single PCLS on a PDMS carrier membrane in a bioreactor. The PCLS was stretched by controlling the pressure applied to the lower chamber of the bioreactor, and the expansion and stretch of the alveoli in the lung tissue were observed and quantified by multiphoton microscopy [42].This “new model” not only provided a new method for observing the changes of the lung in the dynamic lung environment but also opened up a new field for the application of PCLS in disease.

## 4. Application of PCLS Technology in Various Respiratory Inflammatory Diseases

Respiratory diseases are one of the leading causes of morbidity and mortality worldwide, and the extremely high disability rate imposes a huge financial burden on the families of patients [43,44,45,46]. This article lists several classic PCLS inflammatory respiratory disease models, summarizes the applications of PCLS models in diseases, and clearly demonstrates the reliability and utility of PCLS models as an in vitro tool. The detection of inflammatory cells and factors in lung sections in inflammatory diseases of the respiratory system are summarized in Table 1.

### 4.1. PCLSs Can Be Used as a Model of Bacterial Infectious Inflammation and Injury

Acute respiratory inflammation is one of the leading causes of death and disability worldwide and is often caused by bacterial or viral infections [59]. Animal models are often used to study bacterial pneumonia responses. Although the contribution of animal models to basic research is immeasurable, the pathological environment around the human lung is very complex, and the microbial environments in animals and humans are also quite different [60]. Therefore, it is necessary to develop human models to study the pathogenic mechanisms of various pathogens in order to solve the problems caused by species differences. Researchers have begun to use hPCLS model to simulate inflammatory diseases, and some good results have already been achieved.

It was found that in *Mycobacterium abscessus* infections, lesions in the tissue structure or inflammatory infiltration in the alveolar lumen were observed 24–48 h after infection, while animal models required at least 10 to 60 days for lung injury to occur [48,61]. Different types of mice have different infectivity and relative drug resistance, leading to inconsistent results in vivo [62,63]. It was found that in C57BL/6 mice, lymphocytes began to flow into the lungs and clear the bacteria on day 60 after tail vein injection of Mycobacterium abscessus, while lymphocytes near the bacteria were observed 24 h after PCLS infection, showing a mild inflammatory reaction [48].In another study of human lung sections infected with *Staphylococcus aureus*, the bacterium was found to be widely present in the epithelial and mesenchymal regions of hPCLSs and to a lesser extent in a subset of alveolar macrophages. This suggests that the pathogen may not be the target of macrophage phagocytosis during the development of lung inflammation, and the secretory sites of specific toxic substances are not within macrophages [49]. Srijon K et al. established hPCLSs as an infection platform for evaluating *Yersinia pestis* lacking plasminogen activator protease (Pla), which was later found to be mainly phagocytic by alveolar macrophages, and proinflammatory factors in PCLSs were also significantly increased [50]. This technology has evolved into a practical platform for assessing respiratory pathogens infecting the lungs, with exceptionally high application potential and value (Figure 3).

In simulating an injurious inflammatory response, PCLS disease models can be prepared in a specific region with preserved peripheral integrity. These models are simple and reproducible and reproduce the in vivo inflammatory response and the subsequent early repair response well [64]. Acute respiratory distress syndrome (ARDS) is often associated with the activation of systemic inflammatory response, and hypercoagulability in the alveolar cavity is an important indicator of the progression of the inflammatory response [65]. In an ARDS model constructed on a PCLS, it was found that coagulation factor XII (FXII), mainly involved in the intrinsic coagulation pathway, increases rapidly in the early stage of ARDS and can promote the release of inflammatory factors such as interleukins (IL-8, IL-1β, and IL-6), leukemia inhibitory factor (LIF), CXC chemokine ligand-5 (CXCL5), and tumor necrosis factor-alpha (TNF-α) (Figure 3) [51]. Hematopoietic PGD synthase (hPGDS) and its derived prostaglandin D2 (PGD2) are pro-inflammatory mediators produced by immune cells under stimulation and are often expressed in the early stage of respiratory inflammation [66]. Using a PCLS platform, Sonja et al. found that hPGDS and PGD2 are mainly produced by monocytes and macrophages in a time-dependent manner, and inhibiting the production of hPGDS and PGD2 can effectively reduce inflammatory factors in the PCLS medium, providing a new target for the subsequent drug therapy of pulmonary inflammation [59]. In addition, PCLSs can also traumatic inflammation caused by the excessive stretching of the alveoli during the use of mechanical ventilation, and it can provide three-dimensional geometry of the alveolar wall to facilitate the observation of alveolar injury [67].

However, the limitations of PCLSs as an in vitro model were found in a large number of mechanistic studies. It was found that the low number of neutrophils in hPCLS, coupled with a lack of recruited adaptive immune cells, prevented the study of adaptive immune responses throughout inflammatory infection [3]. The researchers then re-injected exogenous neutrophils into hPCLSs to simulate their flow in the body during the inflammatory response to compensate for the lack of circulating neutrophils in PCLSs [60]. Acute lung injury is an inflammatory injury with multiple etiological mechanisms, including acute pancreatitis related to septicemia, blood transfusion-related lung injury, radiation lung injury, etc. [68,69,70]. However, few studies on the mechanisms of acute lung injury have been applied to PCLSs. As an in vitro model, PCLSs are gradually being improved and still have great practical prospects in the study of lung injuries and inflammatory diseases. PCLSs have significant advantages in the study of both infectious and invasive pulmonary inflammation.

### 4.2. PCLS as a Research Tool for Respiratory Viral Infectious Inflammation

Viruses are the leading cause of respiratory diseases and deaths worldwide [71]. Currently, viral pneumonia still lacks specific diagnostic markers, especially the global epidemic outbreak caused by SARS-CoV-2 in 2020, making it imperative for humans to develop good experimental models for breakthrough research on the pathogenesis of the virus and drug development [72]. Compared to animal models and 3D air–liquid interface models in vitro, PCLSs allow all infected cells to be observed more clearly and directly [73].

PCLSs can be utilized as a platform for viral infection in vitro to explore inflammatory mediators and signaling pathways by observing the titer changes of the virus in lung tissues [54]. For instance, to study respiratory tract infections caused by the rhinovirus (RV), transcriptome analysis found that anti-rhinovirus preparations can initiate immune responses on PCLSs, thereby effectively inhibiting pro-inflammatory responses mediated by nuclear factor kappa B (NF-κB) and interferon [74]. The antiviral drug rupintrivir reduces the release of IL-4 and IL-6 inflammatory cytokines in PCLS of in vitro house dust mite (HDM)-sensitized mice, accurately reflecting its therapeutic effect [52]. Adenovirus can induce alveolitis, and the inflammatory cytokine IL-8 was found to be derived from epithelial cells using the immunohistochemical staining of PCLSs, which has positive guiding significance for treatment [53].

Studies conducted using PCLSs have demonstrated that viral infection can replicate itself in the respiratory tract, accompanied by the recruitment of leukocytes, leading to a severe tissue inflammatory response and having a persistent impact on lung function. Limkar et al. detected many virus-induced cytokines highly related to the process of inflammation in a mouse PCLS model of the pneumonia virus infection [54]. With the clearance of the virus in the lungs and the reduction in inflammatory response, the PCLSs of the mice during the recovery period continued to develop severe airway hyper-responsiveness for 45 days [54]. However, some limitations were found in the study, including limited genetic tractability and lack of cellular infiltration and adaptive immunity [60]. PCLS models still need to be improved but are still expected to be applied more in the study of virus-induced inflammatory response mechanism in the future.

### 4.3. Application of PCLS in Classical Chronic Inflammatory Diseases

In chronic inflammatory diseases, PCLS technology has been of more significant benefit. Chronic inflammatory lung diseases include COPD, asthma, idiopathic pulmonary fibrosis (IPF), pulmonary arterial hypertension (PAH), and silicosis [75]. This review details the use of PCLS in chronic inflammatory lung diseases.

#### 4.3.1. Chronic Obstructive Pulmonary Disease

COPD is currently the third leading cause of death globally, with a projected global economic burden of GBP 1.7 trillion by 2030 [76]. COPD is a chronic inflammatory immune response caused by the inhalation of harmful substances [77].

In COPD, the occurrence of inflammatory lesions is closely related to the underlying structure of the lungs. The ideal disease model should reflect the anatomical lesions of the lungs at different stages of inflammation in real time. However, due to many differences in anatomical structure and cellular composition between human and rodent airways, the study of pathological inflammation in airways related in COPD using animal models is limited [78]. For example, rats and mice often do not have airway bronchioles, which are the initial site of central lobular emphysema injury from COPD anatomical lesions, making it difficult to simulate the disease [79]. Basal and non-ciliated luminal cells, which are pools of developing progenitors, were shown to play important roles in repairing the epithelium of the rodent airway. Unlike humans, the inner airway lining of the rat lobule does not have basal cells as an epithelial lining [80]. The basal cells in mouse lungs are confined to the tracheobronchial epithelium, while the basal cells in human lungs extend to the terminal bronchioles. At the same time, the morphology of non-ciliated luminal secretory cells in rodent and human airways are also different between species [81]. These anatomical differences directly affect the reliability of the disease model. Therefore, it is necessary to find optimized in vitro tools [82].

Unlike other disease models, such as rodents, PCLS models can accurately reflect the irreversible pathological changes in chronic progressive inflammatory lung disease and are sensitive to the clinical features of the disease [3]. Characteristic markers of early fibrosis, pro-inflammatory cytokines, and extracellular matrix components collagen I and fibronectin were still observed on PCLSs for up to 5 days [19]. PCLSs are widely used because they closely resemble the disease process in humans. COPD animal models were established via cigarette smoke exposure and elastase induction. The selection of species in smoke exposure models is limited, and the operation technology is difficult [79,83]. However, the short half-life of elastase makes it difficult to restore the disease process in animals for a long time. PCLSs can be used to track characteristic responses in the later stages of inflammation, extending the time the model can be used for disease simulation compared to animal models [21]. Studies have confirmed that the PCLS model can preserve the small airway and its surrounding structure, reflect the relationship between airway structure and function in disease simulation well, and can observe the damage to alveolar repair, the destruction of lung parenchyma, and airway biomechanical changes, which are significant advantages in the simulation of COPD pathophysiology [84,85].

The study found that although PCLSs were isolated from the body’s immune system, they could still produce cytokines to restart the immune response [86]. PCLSs were effective in the evaluation of chronic anti-inflammatory drugs. For example, when studying the pharmacological effects of the inhaled anti-inflammatory dexamethasone, the downregulation of GM-CSF, IL-1β, and IFN-γ factors was observed in the PCLS model, reflecting its therapeutic effect [23]. The PCLS model is helpful in studying the mechanisms of chronic inflammatory immune responses, identifying treatment strategies.

#### 4.3.2. Asthma

Asthma is a chronic heterogeneous inflammatory disease of the airway characterized by airway hyper-responsiveness [87]. With the changes of living environment in the 21st century, the sensitizing factors of asthma are becoming more and more complex, and the prevalence of asthma is increasing worldwide [88]. Early sensitization agent risk assessments to induce airway hyper-responsiveness were mainly carried out in animal experiments, but due to differences in human and animal physiology, there are some limitations of these animal models. Within this context, PCLSs can be used to meet the demand of humans for in vitro research methods in asthma research [89].

Danov et al. directly and clearly observed that cytokines are closely related to allergic asthma in PCLSs. IL-13 can stimulate mucus production and allergic inflammatory markers in human lung and bronchial tissues. Past studies of this factor have relied on animal models, and it was later found that observations on lung sections made the effect of anti-inflammatory treatment more intuitive compared to animal models. Interestingly, airway hyper-responsiveness was only observed in rodent models in response to IL-13 stimulation, not in PCLSs. This is the first evidence that species differences between humans and animals affect the results of asthma research [55]. At present, PCLS models have become the preferred choice for the measurement of narrow-channel hyper-reactive contractions [90]. Donovan et al. used a new dilator, rosiglitazone, to relax the airway in both a mouse model and an in vitro PCLS model of asthma and found that the PCLS model is more intuitive and convenient and the reduction in the inflammatory reaction in vivo is higher [91]. In vitro PCLS models can also be used to observe bronchoconstriction and medium release after airway infection [92]. In asthma, the contraction of the airway smooth muscle (ASM) is the driving force for acute bronchospasm. Using tissue traction microscopy (TTM) to measure the contractile force of airway smooth muscle in a PCLS model, Ram-Mohan et al. found that, compared with lumen area alone, bronchoconstrictor on PCLSs was not only sensitive to stimulation but also had less inter-airway variation, which was convenient to provide spatial information [93]. Agren et al. also observed microscopically that the bronchi in rat PCLSs immediately contracted in a concentration-dependent manner after exposure to high concentrations of ammonia, and they were able to clearly observe the blocking of bronchoconstriction by therapeutic agents. PCLS models have established new protocols for drug therapy and biomarker identification [94].

PCLSs of asthmatic patients can also be used to study the mechanical activation mechanism of pro-remodeling cytokines in the airway, reflecting the feedback between airway mechanics and cytokine activation in asthma and can, thus, better describe the airway’s stress and contractile responses [95].. In conclusion, PCLSs play a reliable role in the research on pathogenesis and treatment of asthma and have broad application prospects.

#### 4.3.3. Other Chronic Inflammatory Lung Diseases

IPF is a chronic, age-related, interstitial pulmonary pneumonia [96]. Hani et al. successfully prepared a PCLS model of IPF using a mixture of pro-fibrotic growth factors, pro-inflammatory cytokines and signaling molecules, and observed early fibrosis features contributing to the evaluation of drug therapy and pathological mechanisms of IPF [19]. It was found that in each patient’s PCLS, the alveolar epithelial cells sustained squamous metaplasia. The reasons for this change include the dysfunction of lung tissue repair, the delayed response after the injury caused by surgical cutting, the addition of FBS in the PCLS culture medium, and the lack of vitamin A and other nutrient elements, which need to be further clarified and solved [31]. Yan Bai et al. showed that PCLSs can preserve intact intrapulmonary arteries and the smooth muscle cells (SMC), mediating their contraction, and can observe arterial contraction in PAH by tracking SMCs contraction and intracellular Ca^2+^ regulatory signals [97]. Other evidence suggests that PCLSs can be used as drug evaluation platforms, which can reflect the improvement in PAH-related endothelial dysfunction, pulmonary inflammation, and pulmonary vascular tone well [57,98,99]. An increasing number of studies have shown that the toxic effects of silica nanoparticles induce chronic inflammation in the lung, leading to silicosis [100]. It was confirmed that alveolar macrophages play an equally important role in vivo [101]. To understand the pathogenesis of silicosis, Falk et al. [58] produced a co-culture system of alveolar epithelial cells and macrophages in a PCLS model. Surprisingly, in early lung injury, macrophages present in the alveoli secreted pro-inflammatory substances such as IL-6, TNF-α, and IL-1β, which instead enhanced the induction of lung inflammation by silica. This co-culture system on PCLSs has become a useful tool in the study of lung injury at the organoid level.

## 5. Optimization and Limitations of PCLS Technology

Although PCLSs are well known as a model to simulate inflammatory diseases, their application still has many limitations:

1. It is challenging to observe the changes in the airway lumen on PCLSs, and it is not convenient to evaluate the therapeutic effect of tracheal administration or determine the source of respiratory tract infection. At present, PCLSs can only be collected and observed in animal models at different times under corresponding conditions, and this method needs to be further optimized [4].

2. PCLSs are isolated from the circulatory system in the body, so researchers are unable to evaluate the migration of cells between the lung, blood, and lymph nodes and are unable to fully replicate the real flow of inflammatory cells and humoral factors in the circulation in the body, which limits the evaluation of adaptive immune inflammatory responses and requires more accurate and comprehensive tracer technology [3,89].

3. The diversity of human genetics can lead to significant differences between individuals, resulting in different slices, and the availability of donors still requires further consideration [89].

4. In the process of practical application, human lung tissue for slicing is not easy to obtain, and a series of standardized processes are required to promote this technology. According to research results, it must be transformed into quantifiable outcome indicators for clinical use, and the projects and indicators of data transformation must be discussed further [21].

5. PCLSs lack a standard and uniform cell medium, and the optimal optimization method has not been found yet. Which elements should be added to the medium to maintain the activity of the PCLSs should be further considered.

## 6. Summary and Prospect

Although PCLS models have some limitations in its application, they are still considered “powerful” compared to other models. With the advancement of technology and times, PCLSs will have increasing “new features”. The use of PCLSs does not only have plenty of room for improvement at the technical level but it also has many further potential applications. With continuous exploration and optimization, we expect that in the near future, we will be able to find a standardized advanced culture technology for PCLS, so that PCLS technology can better bridge the gap between basic research and clinical applications and serve clinical research more effectively.

## Figures and Tables

**Figure 1 bioengineering-09-00767-f001:**
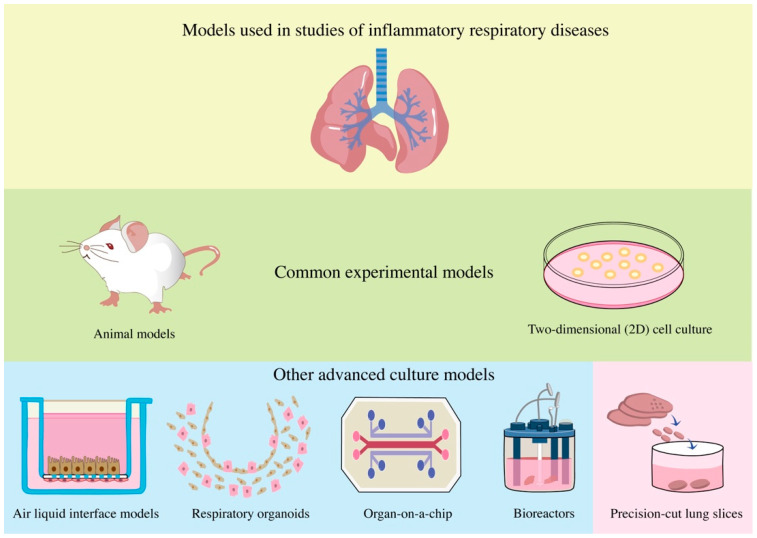
Traditional animal models and 2D cell culture are the common experimental models in studies of inflammatory respiratory diseases. In other advanced culture models, the ALI model can fully reproduce the differentiation of human natural gas tract epithelium with exposure to pathogenic factors in vitro. “Respiratory organoids” can be isolated and amplified from different regions of the mouse and human lungs and then inoculated into 3D cultures to form self-organization. Organ-on-a-chip consists of the air-blood chamber and the attached vacuum channel, which simulates the movement changes of the lung through the stretching of the PDMS membrane and simulates lung function at the organ level. Bioreactors can be seen as vessels for in vitro or ex vivo studies and for the growth of cell and tissue systems, which are more physiologically relevant than traditional static in vitro models. Precision-cut lung slices are thin and uniform 3D lung slices made from human or animal lung tissues. They can reduce the cellular structure and ECM in the native lung environment well while preserving the interaction between different cell types and the ECM.

**Figure 2 bioengineering-09-00767-f002:**
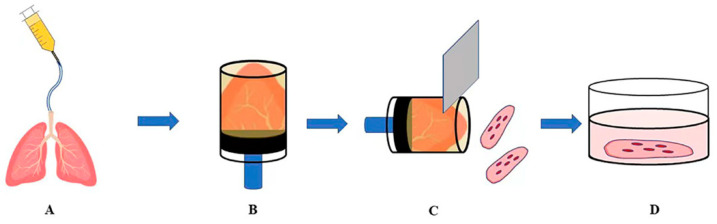
Schematic diagram of a standard preparation of rodent lung slices. (**A**): Fill the lungs with an appropriate volume of 0.5–3% warm agarose solution according to the size of the animal’s lungs to fully inflate the lungs; (**B**): Use a biopsy punch to generate a tissue cylinder with a diameter of 8 mm; (**C**): Use automatic sectioning precise cutting with the machine, usually 100–500 μm thick; (**D**): Place in serum-free MEM with 50 μg/mL gentamicin and 0.25 μg/mL amphotericin at 37 °C, 5% CO_2_ incubation under conditions.

**Figure 3 bioengineering-09-00767-f003:**
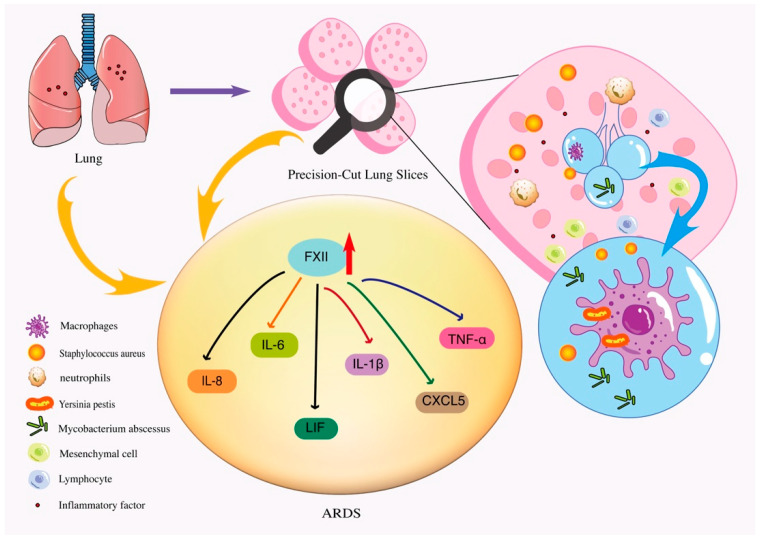
Inflammatory infiltrates in the alveolar space can be observed 24 to 48 h after *Mycobacterium abscessus* infection. *Staphylococcus aureus* is widely present in the epithelial and interstitial areas of hPCLSs, while the number of alveolar macrophages is small, which may not be targeted by macrophages during the development of pneumonia. *Yersinia pestis* is mainly phagocytosed by alveolar macrophages. In the ARDS model constructed on PCLSs, the endogenous coagulation FXII increases rapidly in the early stage of ARDS, which can promote the release of inflammatory factors such as interleukin (IL-8, IL-1β, IL-6), LIF, CXCL5 and TNF-α on PCLSs.

**Table 1 bioengineering-09-00767-t001:** Detection of inflammatory cells and factors in lung sections.

Types of Respiratory Inflammatory Diseases		Method of Preparing the Model	Human/Mouse Lung Slices	Inflammatory Response Evaluation Index	References
Bacterialinfection inflammation	*Pseudomonas aeruginosa*	PCLSs were treated with 5 μg/mL ultrapure Pseudomonas aeruginosa-derived flagellin or 0.1 μg/mL ultrapure LPS derived from *Salmonella R595*	Mouse	Both the chemokines KC and MIP-2 are involved in the early innate immune response by recruiting neutrophils. Dendritic cells release IL1β, IL12, TNF-α, IL-23, IL-6, and IL-8	[47]
	*Mycobacterium abscess*	250 μL DMEM/F12 complete medium was added to PCLSs in a 24-well microplate and inoculated with 1.5 × 10 CFU/plate of *M. abscess* virulent strain L948 (ATCC 19977)	Mouse	Multinucleated cells, epithelioid cells, foamy macrophages, multinucleated giant cells, and early granulomas	[48]
	*Staphylococcus aureus*	*Staphylococcus aureus* cultured in trypsin soybean broth (TSB) lacking sodium pyruvate was incubated with hPCLSs	Human	Large amounts of IL-6 and TNF-α and IL-8 are secreted by human alveolar macrophages	[49]
	*Yersinia pestis*	105 or 107 CO92 wild-type strains or CO92 *Yersinia pestis* strains were infected with hPCLSs in 48-well plates	Human	Alveolar macrophages are the main host cell targets in the early post-infection period, and hPCLSs infection leads to increased levels of TNF-α, IL-6 and IL-8	[50]
ARDS		PCLSs were exposed to FXII or FXIIa, and the expression of inflammatory mediators was assessed using cytokine/chemokine PCR arrays.	Human	Recruitment of neutrophils and macrophages to the injury/infection site initiates the production of inflammatory mediators such as TNF-α, IL-1β, IL-6 and IL-8	[51]
Respiratory viral infection	*Rhinovirus* (RV)	Infect PCLSs with 250 µL of 2 × 10^5^ IU/mL RV, UV-inactivated RV at 33 °C, the optimal replication temperature for RV	Mouse	Increased levels of IFN-α, IFN-β, IFN-γ, MCP-1, TNF-α, IP-10, IL-6, IL-10, and IL-17A were observed on PCLSs	[52]
	*Adenovirus* (Ad)	Lung slices in separate wells were exposed to 1 × 109 pfu/well Ad7 and incubated for 8 h at 37 °C, 5% CO_2_	Human	Neutrophil chemokine IL-8 is mainly derived from epithelial cells, while monocyte and lymphocyte chemokine IP-10 is derived from macrophages and epithelial cells	[53]
	*Pneumonia virus* (PVM)	Intranasal inoculation of 50 µL of mouse pneumonia virus (PVM) strain J3666 and the lungs were harvested to make PCLSs	Mouse	neutrophils recruited to the lung and airway were detected in PCLSs and inflammatory cytokines IL-6, CCL2, and CXCL10were detected in BALF	[54]
Chronic Obstructive Pulmonary Disease (COPD)		Lung slices of mice induced with exogenous elastase to make a COPD model	Mouse	Elastase leads to increased secretion of IL-6, MCP-1, and KC on PCLSs	[20]
Asthma		PCLSs is obtained from the lung lobes of patients undergoing lobectomy for cancer.	Human	IL-13 induced release of proinflammatory cytokines eotaxin-3 and TARC in hPCLS.	[55]
Idiopathic fibrosis (IPF)		A combination of pro-fibrotic growth factors and signaling molecules was used to construct an early fibrosis model in PCLSs from patients without ILD/IPF.	Human	Pro-inflammatory factor IL1B is upregulated on PCLSs	[56]
pulmonary arterial hypertension (PAH)		C57BL/6J mice were intraperitoneally injected with 150 μg/kg lipopolysaccharide (LPS) on the 14th day of gestation, and lung tissue was taken for biopsy on the 28th day	Mouse	The interleukin-1 receptor antagonist down-regulates the pro-inflammatory factor IL-1β	[57]
Silicosis		A co-culture system of alveolar epithelial cells and macrophages was prepared in PCLSs	Mouse	Macrophages secrete IL-6, TNF-α, and IL-1β	[58]

## Data Availability

Not applicable.

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
