# Peer review of "Application of Precision-Cut Lung Slices as an In Vitro Model for Research of Inflammatory Respiratory Diseases"

_bioengineering, 2022, doi:10.3390/bioengineering9120767_

Round 1

Reviewer 1 Report

The manuscript has comprehensively reviewed most of the PCLS literature but some concerns still need to be addressed.

1. English language needs extensive attention and revision

2. Methodology of PCLS preparation and ex-vivo culture needs more details- different methods used, how groups around the world perform agarose filling, in case of smaller lung tissue pieces.

3. Review of the cell culture media used to culture PCLS across various ---studied should be reviewed as well

-----

Others

line 56, use recently instead of Later

line 99-100 please reveiw dfferent instruments used to prepare PCLS such as Krumdeick and vibratome

line 113 Dvornikov et al, 2022, Biorxive use PCLS airways for COPD patients, kindly look into it as well

line 376 The authors showed the cells survive atleast for 2 weeks, but that they only survive for 2 weeks, others have shown transcriptome analysis of PCLS cultured for a month, please review the paper as well (reference # 74) in this context

The American Journal of Pathology, Vol. 192, No. 2, February 2022

The Challenge of Long-Term Cultivation of

Human Precision-Cut Lung Slices

line 380 One of the problems is also the lack of comprehensive optimisation of the cell culture media types used, research commnity still doesnt know which is the best media that can be used to culture PCLS. FACS based analysis of constituent cell types is needed

Author Response

Thank you very much for your comments on the manuscript. According to your suggestion, we have revised the relevant part of the manuscript. Some of your questions have been answered below. Please see the attachment.

Reviewer 2 Report

This is a well written article by Liu et al that explains the scope of using PCLS in research. The stress of the article is on the merits and demerits of PCLS. The details related to the exact experimental methods of using PCLS are limited. The following suggestions may help improve the manuscript better.

Minor suggestions.

1. Please include the use of PCLS in the field of lung development such as alveologenesis by using time lapse imaging with a small description of the details of the method used.

2. A recent publication PMID: 34446466 , has nicely used PCLS to study the role of macrophage derived IL-6 in the pathogenesis of neonatal BPD. This could be used as a reference in addition to driving hoe the message conveyed by the article that PCLS could be pretreated with cytokines and used in experiments

3. There are numerous minor English language corrections required.

4. Mycobacterium abscessus- please write in italics with this being a species name. Same for the words: in vivo or in vitro.

5. The following sentences are unclear  and please rewrite to convey the exact message

Line 165-167: "Acute pathogen infection is often the main cause of pulmonary inflammation, but the mechanism of inflammation is very unclear. In the early days, people used an animal infection model to study the bacterial-induced pneumonia response"

Line 289: "Most animal models are established by inducing inflammation in elastase in vivo."

Line 314: "Past studies of this factor have relied on animal models, and it was later found that observations on lung sections with the effect of anti-inflammatory treatment treatment more intuitive compared to animal models."

Please rewrite this sentence to make the meaning more clear with no grammar mistake.

6. Line 190: "Since this enzyme has a short half-life, it is difficult to obtain in animals. In addition to tracking the early changes in inflammation, PCLS can also be used to follow the characteristic responses in the late stages of inflammation, making it an ideal tool for studying chronic inflammation."

What do the authors intend to convey here? Is Elastase induced in PCLS or not?

Author Response

Thank you for your comments concerning our manuscript. Those comments are valuable and very helpful. We have read through comments carefully and have made corrections. Please see the attachment.

Reviewer 3 Report

In this review by Liu et al, the authors explore the application of precision cut lung slices (PCLS) within the context of inflammatory respiratory diseases. The authors introduce the current methodology used to study disease and describe how PCLS compares as an in vitro method. The authors should be commended for their work outlining the literature. The following are a few suggestions which could help strengthen the manuscript. 

1.      In the introduction, the authors provide a series of alternative models and discuss their relative pros and cons, which is helpful for contextualizing the PCLS. This section could be aided by a figure to really illustrate how many other methods there are and what they consist of. Additionally, the discussion of organoids and microfluidic chips could have a few more introductory sentences to describe how they are made to better understand what their limitations are. Also, as the following section details the preparation and storage of the PCLS in much more detail, the authors don’t need a full outline in the previous section of what PCLS is, but there could be a bit more explanation when the lung slices are first introduced. For example, on page 1, on line 70, the authors could briefly describe that lung slices are derived from cutting agarose-filled lungs using automated microtomes. This could help orient a reader with less PCLS background.    

2.      In section 2, the authors describe one method to creating rodent PCLS and say that there are various methods. It could be informative to a reader to know what other ways there are, even very briefly. The authors could also describe relative advantages and disadvantages to making certain choices while preparing the PCLS. The authors touch on this in a later section, but this section would be an excellent point at which to bring up how culturing conditions could affect PCLS and what choices are made throughout the literature.

3.      The authors introduce cryopreservation on line 117 on page 3 but do not describe what it consists of. They also write on lines 121 that “PCLS encapsulated with hydrogel biomaterials can maintain he ability to grow for at least 21 days” but more detail would clarify exactly what is meant by this. What did that paper measure as output? How were the PCLS in this case different from other methods tested in the paper and what were the results?

4.      When discussing cryopreservation at the end of section 2 on page 3, Liu et al write that “the airway response still presents” but do not detail what response is meant by this.

5.      Similarly the authors could provide more detail on page 3 line 134 when they write that PCLS can “restore the tumor microenvironment in patients”. More details on the findings of these papers which Liu et al are reference would help the reader understand the potential of PCLS. The authors raise some interesting points about the PCLS and how they are used to understand ultrastructural changes, but remain still vague when they write “the application of fluorescent markers and probes” on lines 148-150 of page 4. This is an opportunity for the authors to outline what has been studied so far to engage the reader in understanding how powerful the generation of PCLS can be.

6.      Table 1 is very logically laid out and does a good job of organizing the types of diseases by type, but is a bit vague when describing the outcomes of the studies. The table could be a good place for readers to refer to a summary of major findings, but currently only lists the cytokines and cell types that were studied in certain publications without referencing what the findings actually were. The authors could, for example, divide out into two columns those factors which were found upregulated vs downregulated or outline what was actually determined about the cell types that they list.

7.      On page 5 lines 174-175, the authors could more clearly outline what the findings of the PCLS in terms of mycobacterium infections were compared to animal models and how those animal models were established. It is a very interesting finding that infection was observed after 24-48 hours compared to 10-60 days and would be worth commenting on why this is the case.

8.      The authors could expand more when they say on page 7 lines 255 that the hydrogel can expand the lifespan of the PCLS. This is the same as the previous comment #3.

9.      If there are no limitations to figures or tables, the authors could add in a table for sections 4.3.1 and 4.3.2 to detail the findings in COPD and asthma that are unique to PCLS. This would serve to highlight how this model differs from others in establishing disease pathogenesis.

10.  On page 8 lines 286-287, when the authors refer to a more accurate capturing of changes in PCLS, do they mean in comparison to how other animal models have depicted COPD? Could the authors elaborate in what way the other models are deficient and why the PCLS are able to capture these changes better, if that information is available?

11.  On page 9 lines 315-316, it is unclear what the clause “It was later found that observations on lung sections with the effect of anti-inflammatory treatment ore intuitive compared to animal models” means. Could the authors please rephrase?

12.  The authors write on page 9 lines 333-334 that “PCLS has established new protocols for drug therapy and biomarker identification” without describing at all what these new methods or protocols entail. 

13.  What does it mean on page 9 lines 360-363 to have a “persistent loss of a small percentage of cells” along with a “persistent increase in chemotactic cells”. Which cells are lost? How was this observed and what was the impact?

14.  The authors write on page 10 line 395 that “a series of standardized processes are required to promote” PCLS. This would be a good opportunity within the scope of the review for the authors to suggest what exactly needs to be standardized. They have not really detailed what is different between the tissue slicing methods to then delve on which aspects need to be standardized. For a reader who does not share the same expertise in PCLS, this section would be an opportunity to understand the growth of the field and its potential future.

15.  The authors bring up acute lung injury within the summary in section 6 on page 10, but that discussion of how ALI is rarely applied in PCLS seems better suited to a previous section rather than a conclusive summary.

Minor Points:

1.      Missing an “and” on line 30 before chronic obstructive pulmonary disease

2.      Phrasing of “can better restore the spatial structure of the lung from the perspective of three-dimensional” on page 2 lines 47-48 which could be instead: “From a three-dimensional perspective, the air-liquid interface (ALI) model can better resolve the spatial structure of the lung…”

3.      On page 2 lines 90-97 describing how PCLS are prepared, the past and present tenses are mixed. It would make more sense to consistently use one and present tense appears to be a more appropriate choice in this context.

4.      The phrase “waterfall-type” on line 201 on page 6 is not a common description for the inflammation in acute respiratory distress syndrome and other wording could be used to make the concept more clear.

5.      The abbreviation PGD is used on line 208 on page 6 without previously being defined.

6.      The phrasing on page 8 lines 305-306 is slightly awkward. “With the change of people’s living environment and way in the 21st century” could perhaps be better rephrased as “With changes in the living environment and habits in the 21st century” or similar.

Overall, the authors provide a useful summary of inflammatory disease within PCLS. With greater detail and more clarification, the work would be helpful in providing readers an understanding of what has already been established in PCLS and what could be gained from further work with the methodology.

Author Response

Thanks very much for your hard work and professional comments, and we are really sorry for our inaccuracy. According to your suggestions, we have made some changes in the article  and we have replied to the suggestions one by one. Please see the attachment.

Reviewer 4 Report

This review discusses the use of precision cut lung slices (PCLS) derived from human or animal tissue for the study of respiratory diseases. The topic is interesting and the paper addresses fairly the pros and cons of the procedure compared to other in vitro and in vivo techniques and describes its applications to the study of different lung disorders. The reference list appears to be extensive.  

A major limitation of this manuscript is the lack of experimental details (including a critical evaluation) for such a procedure, requiring fine-tuned culture conditions and technical skills. This section should be included. The language should be improved. 

Additional comments 

Introduction, first paragraph: a whole sentence is repeated. 

Lines 157-158. This sentence suggests a relationship between the financial burden of the disease and its pathogenesis, an awkward hypothesis.  

Lines 383-385. This statement is unclear. 

Lines 399-400. The enthusiasm of the authors for this technique is remarkable. However, stating that its development will involve the whole mankind appears excessive.

Author Response

Thank you very much for your letter. According to your suggestions, we have made some changes in the article  and we have replied to the  suggestions one by one. Please see the attachment.

Round 2

Reviewer 1 Report

The manuscript has improved majorly since the last revision.

If possible it would be nice to highlight that (in Line 139 to 141) in hPCLS system, where the lung is not availble with bronchus, the tissue ressection piece is inflated by random poking of agaorsee into the tissue till it is sufficiently inflammed (Khan et al., 2021 ERJ).

--> Please highlight the work of Khan et al 2021 ERJ, where authors show that PCLS can maintain many cell type markers for 2 weeks in culture without using FCS. This work has analysed lung slices on proteomic and metabolic levels.

Author Response

Thanks very much for your hard work and professional comments. We have made some changes in the article  and we have replied to the reviewer 1's suggestions one by one. Please see the attachment.

Reviewer 3 Report

The authors have respoonded sufficiently to the comments

Author Response

Thank you very much for your comments. These suggestions give us great inspiration and help us to improve this article better.